# A complete tool set for molecular QTL discovery and analysis

Olivier Delaneau[1,2,3], Halit Ongen[1,2,3], Andrew A. Brown[1,2,3], Alexandre Fort[1], Nikolaos I. Panousis[1,2,3] & Emmanouil T. Dermitzakis[1,2,3]

Population scale studies combining genetic information with molecular phenotypes (for example, gene expression) have become a standard to dissect the effects of genetic variants onto organismal phenotypes. These kinds of data sets require powerful, fast and versatile methods able to discover molecular Quantitative Trait Loci (molQTL). Here we propose such a solution, QTLtools, a modular framework that contains multiple new and well-established methods to prepare the data, to discover proximal and distal molQTLs and, finally, to integrate them with GWAS variants and functional annotations of the genome. We demonstrate its utility by performing a complete expression QTL study in a few easy-to-perform steps. QTLtools is open source and available at https://qtltools.github.io/qtltools/.

[1] Department of Genetic Medicine and Development, University of Geneva, 1 Michel Servet, Geneva CH1211, Switzerland. [2] Swiss Institute of Bioinformatics, University of Geneva, 1 Michel Servet, Geneva CH1211, Switzerland. [3] Institute of Genetics and Genomics in Geneva, University of Geneva, 1 Michel Servet, Geneva CH1211, Switzerland. Correspondence and requests for materials should be addressed to O.D. (email: olivier.delaneau@gmail.com) or to E.T.D. (email: emmanouil.dermitzakis@unige.ch).

To increase the explanatory power of genome-wide association studies (GWAS), many genetic studies now routinely combine genetic information with one or multiple molecular phenotypes such as gene expression[1–3], protein abundance[4], metabolomics[5], methylation[6] and chromatin activity[7]. This makes the discovery of molecular Quantitative Trait Loci (molQTL) possible; a key step towards better understanding the effects of genetic variants on the cellular machinery and eventually on organismal phenotypes. In practice, this requires analysing data sets comprising of millions of genetic variants and thousands of molecular phenotypes measured on a population scale; a design that aims to perform orders of magnitude more association tests than in a standard GWAS, which prevents the use of standard tools designed to handle only few phenotypes[8,9]. To face this computational and statistical challenge, there is a clear need of computational methods that are (i) powerful enough to handle the multiple testing problem, (ii) fast enough to easily process large amounts of data in reasonable running times and (iii) versatile enough to adapt to new data sets as they are being generated. Here, we present such an integrated framework, called QTLtools, which allows users to transform raw sequence data into collections of molQTLs in a few easy-to-perform steps, all based on powerful methods that either match or improve those employed in large scale reference studies such as Geuvadis[1] and GTEx[10].

QTLtools is a modular framework designed to accommodate new analysis modules as they are being developed by our group or the scientific community. In its current state, QTLtools performs multiple key tasks (Fig. 1) such as checking the quality of the sequence data, checking that sequence and genotype data match, quantifying and stratifying individuals using molecular phenotypes, discovering proximal or distal molQTLs and integrating them with functional annotations or GWAS data. To demonstrate the utility of this new tool with real data, we used it to perform a complete expression QTL (eQTL) study for 358 European samples where genotype and expression data were generated as part of the 1,000 Genomes[11] and Geuvadis[1] projects (Supplementary Data 1).

## Results

**Controlling the quality of the sequence data**. To control the quality of the sequence data, QTLtools proposes two complementary approaches. First, it can measure the proportions of reads (i) mapping to the reference genome and (ii) falling within an annotation of interest (Supplementary Note 1), such as GENCODE for RNA-seq[12]. Second, it can ensure that the sequence data matches the corresponding genotype data; the opposite being an evidence of sample mislabelling[13]. To achieve this, QTLtools measures concordance between genotypes and sequencing reads, separately for heterozygous and homozygous genotypes (Supplementary Note 2). Low values in any of the two measures indicate problems such as sample mislabelling, contamination or amplification biases (Supplementary Fig. 1). When performed on Geuvadis, these two approaches demonstrate the high quality of the RNA-seq data (Supplementary Fig. 2) and the good match with available genotype data (Supplementary Fig. 3).

**Quantifying gene expression**. To quantify gene expression, QTLtools counts the number of sequencing reads overlapping a set of genomic features (for example, exons) listed in a given annotation file (Supplementary Note 3). We quantified both exon and gene expression levels in all 358 Geuvadis samples using this approach and find 22,147 genes with non-zero quantifications in more than half of the samples (Supplementary Fig. 4). Then,

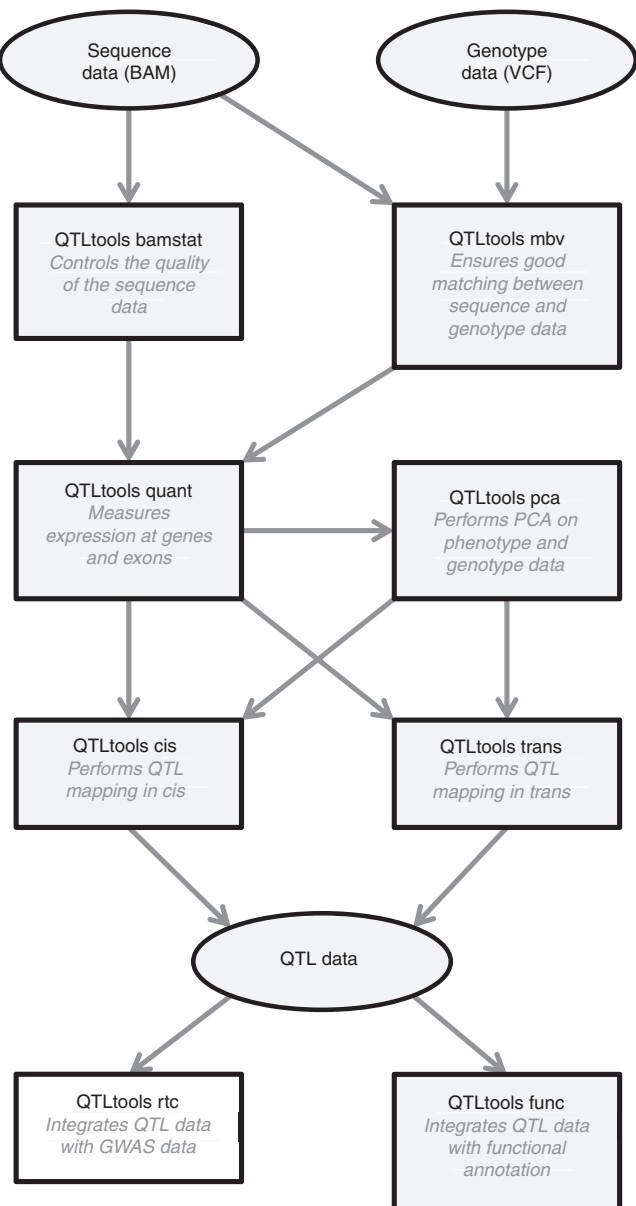

**Figure 1 | Flow chart of the main QTLtools functionalities.** This represents how the various functionalities of QTLtools can be combined to go from the raw sequence and genotype data to collections of molecular QTLs which can then be integrated with both GWAS data and functional annotations. Data is represented with ovals and tasks with boxes in which the name of the mode is shown in bold black with a short description of what it does.

we run principal component analysis (PCA) on these quantifications, as implemented in QTLtools (Supplementary Note 4), to capture any stratification in the sequence data or in the genotype data. In the Geuvadis data, we did not observe any unexpected clusters in the expression data or in the genotype data (Supplementary Fig. 5) and used the resulting weights on the first few principal components as latent variables to increase discovery power of any downstream association testing (Supplementary Note 5).

**Mapping proximal molQTLs**. A core task of QTLtools is to discover proximal (that is, *cis*-acting) molQTLs. To do so, it extends the QTL mapping method introduced by FastQTL[14] and offers multiple key improvements that make this step fast and

easy-to-perform. First, it uses a permutation scheme that needs a relatively small number of permutations to adjust nominal *P* values for multiple testing (see Methods section and Supplementary Fig. 6). As a consequence, the whole-Geuvadis eQTL analysis can be performed in short running times ($\sim$32 CPU hours) which has previously been proved to be an order of magnitude faster than a widely used tool, Matrix eQTL[15] and provides adjusted *P* values without any lower bounds (Supplementary Fig. 7). The running times are actually so small that it becomes possible to process rapidly massive data sets such as the GTEx v6p study[16] (7,051 samples in $\sim$870 CPU hours; Supplementary Fig. 8) and to repeat the whole analysis multiple times across different sets of quantifications, covariates and QC filters to determine the optimal configuration which maximizes the number of discoveries (Supplementary Figs 9 and 10). In addition, QTLtools also provides ways to easily extract subsets of data and therefore facilitate detailed inspection of particular eQTLs (Supplementary Fig. 11).

**Mapping proximal molQTLs for groups of phenotypes**. As multiple molecular phenotypes can belong to higher order biological entities, for example exons lying within genes or histone modification peaks which form larger variable chromatin modules (VCMs[7]), we also implemented two methods to maximize the discoveries in such particular cases (Methods section). Specifically, QTLtools can either (i) aggregate multiple phenotypes in a given group into a single phenotype via PCA or (ii) directly use all individual phenotypes in an extended permutation scheme that accounts for their number and correlation structure. In our experiments, the permutation-based approach seems to outperform the PCA-based approach in terms of number of discoveries in the two data sets we tested (Fig. 2a, Supplementary Data 2, Supplementary Fig. 12). In Geuvadis, the permutation-based approach is able to discover an additional set of $\sim$1,056 eQTLs compared to the standard gene-level quantifications, most of them being for genes containing many exons (Supplementary Fig. 13).

**Mapping proximal molQTLs using conditional analysis**. Furthermore, QTLtools can also perform conditional analysis to discover multiple proximal molQTLs with independent effects on a molecular phenotype. To do so, it first uses permutations to derive a nominal *P* value threshold per molecular phenotype that varies and reflects the number of independent tests per *cis*-window. Then, it uses a forward–backward stepwise regression to (i) learn the number of independent signals per phenotype, (ii) determine the best candidate variant per signal and (iii) assign all significant hits to the independent signal they relate to (Methods section). We applied this conditional analysis on Geuvadis and discovered that $\sim$38% of the significant genes have actually more than one eQTL (Fig. 2b); some have up to six independent eQTLs (Fig. 2c). Interestingly, we also find that combining the conditional analysis with the phenotype grouping approach described above could help to discover even more signals (Fig. 2b,c). The new discoveries resulting from theses analyses in Geuvadis have high replication rates within an independent data set (GTEx[10]) suggesting that these are genuine discoveries (Supplementary Note 6, Supplementary Fig. 14).

**Mapping distal molQTLs**. Beyond mapping proximal molQTLs, QTLtools also includes methods to discover distal (that is, *trans*-acting) molQTLs. The first method we implemented relies on permuting all phenotypes together to draw from the null distribution of associations while preserving the correlation structure within genotype and phenotype data intact (Methods

section). By repeating this permutation scheme multiple times (for example, 100 times in our experiments), we can obtain an empirically calibrated Quantile–Quantile plot that properly shows signal enrichment (Fig. 2d) and can estimate the false discovery rate (FDR) for all the most significant associations: in Geuvadis, we could find 52 genes with at least one significant signal in *trans* at 5% FDR. Given that this full permutation scheme is computationally intensive ($\sim$450 CPU hours for 100 permutations), we also designed an approximation of this process that gives reasonably close FDR estimates while being multiple orders of magnitude faster ($\sim$7 CPU hours; Methods section). Given that the whole genome is effectively tested for each phenotype, we quickly build a null distribution of associations for a single phenotype by permutations. We then use this null distribution to adjust each nominal *P* value for the number of variants being tested and then use standard FDR methods[17] on the resulting set of adjusted *P* values to correct for the multiple phenotypes being tested. In practice, this approach can be seen as an extension of the mapping strategy we use in *cis* for *trans* analysis, and gives FDR estimates that are close to those obtained with the full permutation pass (Supplementary Fig. 15) while being much faster to obtain ($\sim$64 times faster in our experiments).

**Integrating molQTLs with GWAS and functional data**. Finally, we also implemented multiple methods to integrate collections of molQTLs with two types of external data: functional genome annotations and GWAS results. First, QTLtools can estimate if a molQTL and a variant of interest (typically a GWAS hit) pinpoint the same underlying functional variant. To do so, it uses regulatory trait concordance[18] (Supplementary Note 7); a sophisticated conditional analysis scheme designed to account for linkage disequilibrium as a confounding factor when co-localizing molQTLs and GWAS hits. This can be used, for instance, to determine the subset of GWAS hits that are likely mediated by molQTLs; a useful piece of information to understand the function of GWAS hits. When applied on Geuvadis and the NHGRI-EBI GWAS catalogue[19], we estimated to which extent the disease associated variants reported in this catalogue overlap with eQTLs for lymphoblastoid cell lines (Supplementary Fig. 16). Alternatively, QTLtools can also look at the overlap between molQTLs and functional annotations such as those provided by ENCODE[12]. Specifically, it can compute the density of annotations around molQTL locations and, when they do overlap, estimate if it is more often than what is expected by chance (Methods section). This allows the distribution of functional annotations around molQTLs to be inspected visually (Fig. 2e) and statistically (Fig. 2f). When using this on the various sets of eQTLs we have discovered so far, we find that they tend to fall within transcription factor binding sites and open chromatin regions (Fig. 2f), in line with previous knowledge on eQTLs[1].

**Computational efficiency**. All functionality described above has been implemented in C++ for high performance and in a modular way to facilitate future implementation of additional functionalities by the community. In practice, this allows all the experiments described above to be run in a relatively short time (Supplementary Table 1); the full set of analyses described above were completed in $\sim$1,327 CPU hours ( = $\sim$55 CPU days). In addition, QTLtools has been designed so that the computational load can be easily distributed across the multiple CPU cores that are typically available on a compute cluster. The tasks run on individual samples (for example, QC the sequence data) are simple to parallelize as one compute job per individual. For population-based tasks, such as QTL mapping, the input data is

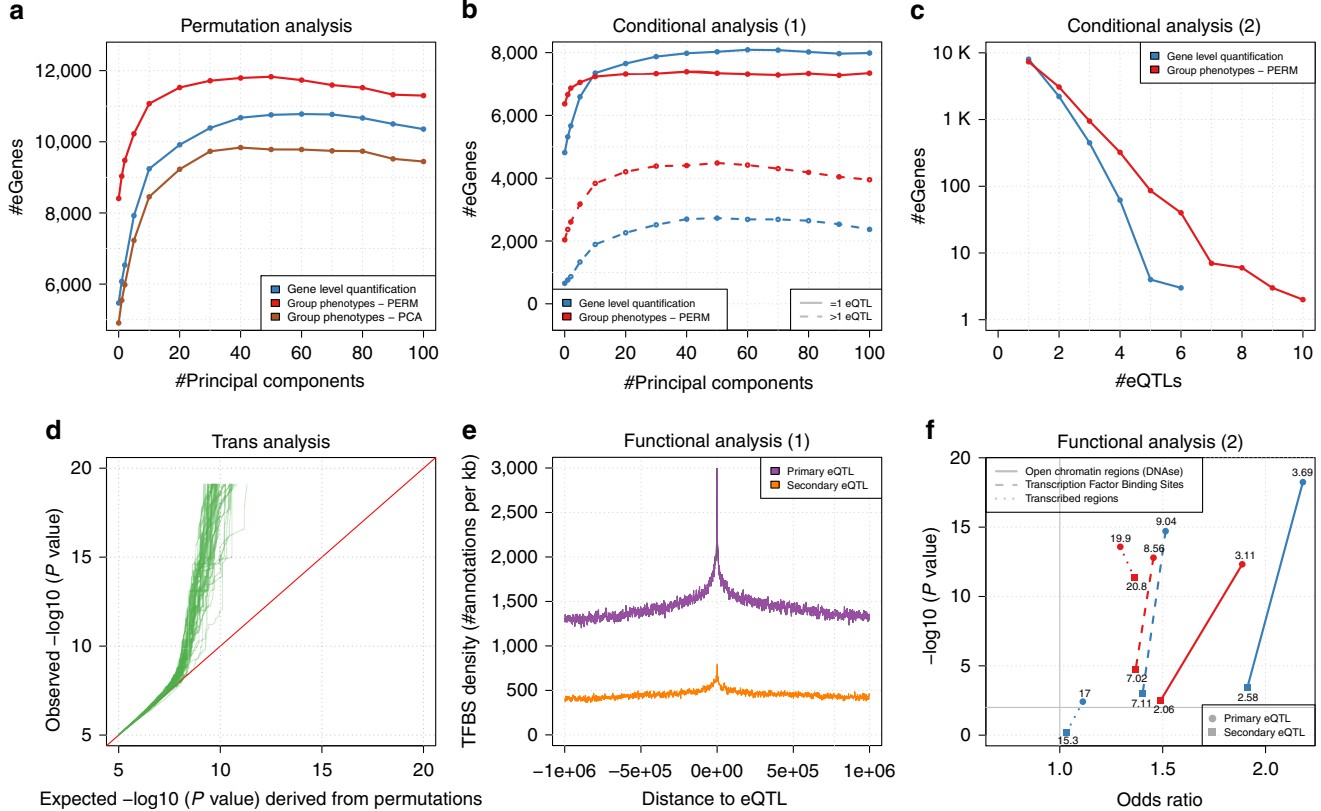

**Figure 2 | Outcome of multiple key analyses on Geuvadis.** (**a**) The number of eGenes discovered (*y* axis) as a function of the number of Principal Components (*x* axis) used to correct for technical variance for three different ways of aggregating signal at multiple exons: at the quantification level (in blue) or at the QTL mapping level by using either the extended permutation scheme (in red) or PCA (in brown). (**b**) The numbers of eGenes (*y* axis) with a unique eQTL (solid lines) or multiple eQTLs (dotted lines) as a function of the number of principal components (*x* axis) used to correct for technical variance. This is shown for two approaches for aggregating the signal at multiple exons: at the quantification level (in blue) or at the QTL mapping level by using the extended permutation scheme (in red). (**c**) The number of eGenes on a log scale (*y* axis) as a function of the number of independent eQTLs discovered for those (*x* axis). This is again shown for two different approaches for aggregating the signal at multiple exons. (**d**) A Quantile–Quantile plot produced from a *trans*-QTL analysis on Geuvadis. Each green solid line compares the *P* values of associations of the original gene expression data to those obtained from a permuted data set. In total, 100 permutations have been performed, resulting in 100 green lines. (**e**) The density of transcription factor binding sites (TFBS) as their number per kb around the positions of two types of eQTLs shown in **b** (primary and secondary, gene-level quantification). (**f**) The enrichments of the four types of eQTLs shown in **b** (primary versus secondary, gene quantification versus phenotype grouping) within three types of functional annotations (Methods section). The odd ratios and the  − log10 of the enrichment *P* values are shown on the *x* axis and *y* axis, respectively. The percentages of eQTLs falling within these annotations are shown next to the corresponding points.

automatically split into small genomic chunks that are then run conveniently and independently on distinct CPU cores.

## Discussion

Population scale studies combining genetic variation and molecular phenotypes have become a standard to detect molecular QTLs. This requires multiple computational steps to go from the raw sequence and genotype data to collections of molecular QTLs. So far, this can be done using multiple tools that are often hard to combine and/or adapt to the amount of data involved. We propose in this paper, QTLtools, a software package that integrates all functionalities required to easily and rapidly perform this task. It includes multiple new and powerful statistical methods to prepare and control the quality of the data, to map proximal and distal QTLs and to integrate those with GWAS results and functional annotations. It also offers a unique framework for the community to develop further additional methods or alternative to the ones already included, so that molecular QTL analysis can be more seamless among laboratories. By its integrative design and efficient implementation, QTLtools dramatically decreases the time needed to set up

and run the various analysis pipelines traditionally needed by molecular QTL studies, freeing researchers to spend more effort on the interpretation and validation of their results.

## Methods

**Mapping proximal molQTLs using permutations.** Mapping proximal molecular QTL consists of finding statistically significant associations between molecular phenotypes and nearby genetic variants; a task commonly undertaken using linear regressions[1,10,14]. In practice, this requires millions of association tests to scan all possible phenotype-variant pairs in *cis* (that is, variants located within a specific window around a phenotype), resulting in millions of nominal *P* values. Due to the large number of tests performed per molecular phenotype, multiple testing has to be accounted for to assess the significance of any discovered candidate molQTL. A first naive solution to this problem is to correct the nominal *P* values for the number of tested variants using the Bonferroni method. However, due to the specific and highly variable nature of each genomic region being tested in terms of allele frequency and linkage disequilibrium, the Bonferroni method usually proves to be overly stringent and results in many false negatives. To overcome this issue, a commonly adopted approach is to analyse thousands of permuted data sets for each phenotype to empirically characterize the null distribution of associations (that is, the distribution of *P* values expected under the null hypothesis of no associations). Then, we can easily assess how likely an observed association obtained in the nominal pass originates from the null, resulting in an adjusted *P* value. In practice, thousands of permutations are required in this context and therefore fast methods able to absorb such substantial computational loads in

reasonable running times are needed. FastQTL has recently emerged as a good candidate for this task by proposing a fast and efficient permutation scheme in which the null distribution of associations for a phenotype is modelled using a beta distribution[14]. This allows approximating the tail of the null distribution relatively well using only few permutations and also accurately estimating adjusted $P$ values at any significance level in short running times. In the original FastQTL paper, it has been shown that running 1,000 permutations gives accurate adjusted $P$ values while being $\sim 17$ times faster than when implementing the standard permutation scheme with MatrixeQTL running on a BLAS-optimized R version[15]. In QTLtools, we use exactly the same approach than in FastQTL: we approximate the permutation outcome with a beta distribution. We run this method on Geuvadis using 1,000 permutations and a *cis*-window of 1 Mb. And since the eQTL mapping is quick and easy, we repeated the whole-mapping pass multiple times across multiple conditions. Specifically, we repeated the whole-Geuvadis analysis across multiple missing data proportion filters (that is, %genes with RPKM = 0 between 0 and 100; Supplementary Fig. 10) and numbers of expression-derived Principal Components (that is, PCs between 0 and 100; Supplementary Fig. 9). We therefore determine that the optimal configuration to maximize the number of discoveries relies on filtering out genes with more than 50% of the samples with non-zero quantifications and using 50 expression-based PCs as covariates. In all downstream analyses, we used this configuration when not specified otherwise.

**Mapping proximal QTLs for groups of phenotypes.** It is common that some kinds of molecular phenotypes may belong to higher order biological entities. For instance, a given gene often contains multiple exons. Similarly, nearby regulatory elements may cooperate within some module structures such as VCM[7] or topologically associated domains[20]. To map molecular QTLs at the level of these higher order biological entities, we need methods able to properly combine information at all the multiple molecular phenotypes they contain. In the context of genes, this has traditionally been done at the quantification level: read counts at multiple exons are summed up to get gene-level quantifications that are then used to discover gene-level eQTL. In QTLtools, we introduced two approaches to combine locally multiple molecular phenotypes belonging to a given group. First, we extended the permutation scheme described above to deal with each group of phenotype independently. Assuming that a phenotypic group $P$ (for example, gene) contains $M$ phenotypes (for example, exons) and that the corresponding *cis*-window $G$ contains $L$ genetic variants, QTLtools proceeds as follows:

(1) All $M{\times}L$ possible variant-phenotype pairs are tested using linear regressions. The pair with the smallest nominal $P$ value is stored as best candidate QTL for this group of phenotype.
(2) Permute simultaneously all phenotypes in $P$ using the same random number sequence. As a result, the inner correlation structure within both $P$ and $G$ remains completely unchanged, while the correlation in between $P$ and $G$ is broken.
(3) Draw from the null distribution of association between $P$ and $G$ by scanning all $M{\times}L$ possible variant-phenotype pairs in the permuted data set and by retaining the best association.
(4) Build empirically the null distribution of association between $G$ and $P$ by repeating the steps (2) and (3) as many times as needed (typically 1,000 times is enough).
(5) Fit a beta distribution on this empirically defined null distribution using expectation-maximization[14]. This effectively makes the null distribution continuous.
(6) Adjust the nominal $P$ value of the best pair obtained in step (1) using the fitted beta distribution.
(7) Repeat step (1) to (6) for all groups of phenotypes to get a candidate QTL together with an adjusted $P$ value of association for each.
(8) Determine all significant QTLs at a given FDR (typically 5%) using a FDR procedure such as Storey–Tibshirani implemented in $R/q$ value[17] on the adjusted $P$ values.

Note that this permutation scheme corrects for both the number of genetic variants and the number of molecular phenotypes being tested while properly accounting for their inner correlation structure. As a consequence, when the beta distribution is fitted in step (5), we get an estimate of the effective number of independent tests corresponding to the actual $M{\times}L$ tests we performed. Alternatively to this extended permutation scheme, we also implemented an approach based on dimensionality reduction. This has been previously used to discover QTLs for VCMs from single-ChIP-seq peak quantifications[7]. Here, a PCA is first performed on the $M$ phenotypes and the loadings on the first PC are used as a quantification vector for the entire group of phenotypes. We can then perform the standard mapping approach implemented in QTLtools to discover a QTL for $P$. We applied these two approaches on Geuvadis to discover gene-level eQTL from exonic quantifications and compared them with the standard gene-level quantifications. We find that the largest number of eQTL is obtained with the extended permutation scheme and the smallest with the PCA-based approach; the gene-level quantifications lying in between. The boost provided by the extended permutation scheme is really appreciable since we get an additional set of 1,019 eQTLs that the gene-level quantification is unable to discover (Fig. 2a). Of note,

it really helps to discover eQTL for genes having a high number of exons (Supplementary Fig. 13). In addition to this, we also applied both approaches on the histone modification data to discover vcmQTL and find similar results (Supplementary Fig. 12). Despite the lower performance of the PCA-based approach in this context, we decided to keep it in QTLtools since we believe it can still be useful in a different context; such as for instance when we are more interested in capturing instead the common trend between multiple phenotypes within a group.

**Mapping proximal molQTLs using conditional analysis.** The two mapping approaches above only report a single candidate QTL per phenotype or group of phenotypes. In some cases, this limitation may reduce significantly the number of discoveries. For example, it is relatively frequent that expression for a given gene is affected by multiple proximal eQTLs[1]. A well-established approach to discover multiple QTLs with independent effects on a given phenotype relies on conditional analysis: new discoveries are made by conditioning on previous ones. In QTLtools, we implemented a conditional analysis scheme based on stepwise linear regression that is fast, accounts for multiple testing and automatically learns the number of independent signals per phenotype. Specifically, we implemented it as follows for both grouped and ungrouped phenotypes:

(1) *Initialization.* We determine a nominal $P$ value threshold of significance on a per-phenotype basis. To do so, we first perform a permutation pass as described above which gives us an adjusted $P$ value per phenotype (or group of phenotypes) together with its most likely beta parameter values. Next, we determine the adjusted $P$ value threshold corresponding to the targeted FDR level (for example, 5% FDR) and feed the beta quantile function (for example, $R/q$ beta) with it to get a specific nominal $P$ value threshold for each phenotype. Here, the beta quantile function allows us to use the Beta distribution in a reversed way: from adjusted $P$ value to nominal $P$ value. Note that the resulting nominal $P$ value thresholds vary from one phenotype to the other depending on the complexity of the *cis* regions being tested and the effective number of independent tests they encapsulate.
(2) *Forward pass.* We next learn the number of independent signals per phenotype using stepwise regressions with forward variable selection. More specifically, we start from the original phenotype quantifications and search for the variant in *cis* with the strongest association. When the corresponding nominal $P$ value of association is below the threshold defined in step (1), we store the variant as additional and independent discovery and residualize its genotypes out from the phenotype quantifications. We then repeat these two steps until no more significant discovery is made: this immediately gives us the number of independent molQTLs together with a best candidate variant for each.
(3) *Backward pass.* Finally, we try to assign nearby genetic variants to the various independent signals we discovered in step (2). To do so, we define a linear regression model that contains all candidate QTLs discovered so far in the forward pass: $P = Q_1 + \ldots + Q_i + \ldots + Q_R$ where $R$ is the number of independent signals and $\{Q_1, \ldots, Q_i, \ldots, Q_R\}$ are the corresponding best molQTL candidates. Then, we test all possible hypotheses by fitting this model $R{\times}(L{-}R)$ times each time fixing $\{ Q_1, \ldots, Q_{i-1}, Q_{i+1},\ldots, Q_R \}$ and setting $Q_i$ as another variant in *cis* ($L{-}R$ variants in *cis* not being a candidate molQTL times $R$ independent signals). We then end up with a vector of $R$ nominal $P$ values for each variant in *cis* which allows us to determine the signal the variant belongs to by simply finding the smallest $P$ value in this vector and comparing it to the significance threshold obtained in step (1).

**Mapping distal molQTLs.** Another common problem in the field of QTL discovery relates to mapping distal QTLs (that is, *trans*-QTL). This presents multiple computational and statistical challenges related to multiple testing, computational feasibility and confounding factors such as read misalignment, gene homology or incorrect gene location. In the context of this work; we only address two particular problems: how to correct for multiple testing and how to perform this analysis in reasonable running times. We solved this problem by testing all possible phenotype-variant pairs for association excluding all those in *cis* (that is, implying that the phenotype and the variant cannot be proximal, typically $< 5$ Mb) using linear regressions with high computational performance as we do for *cis* mapping. In practice, we manage to perform $\sim 1.3$ M linear regressions per second for 358 individuals on an AMD Opteron(tm) Processor 6,174 at 2.2 GHz. To minimize the RAM usage, the phenotype data is stored in memory and the genotype data streamed as we move along the genome and tested against all phenotypes at once. To minimize the size of the output files, we only report detailed information for associations below a given threshold (typically $10^{-5}$ for nominal $P$ values); all those above are simply binned to have an idea of the overall $P$ value distribution. Once the nominal pass done, we correct for multiple testing using one of these two approaches:

(1) *Full permutation scheme.* We permute all phenotypes using the same random number sequence to preserve the correlation structure unchanged. By doing so, the only association we actually break in the data is between the genotype and the phenotype data. Then, we proceed with a standard association scan identical to the one used in the nominal pass. In practice, we repeat this for 100

permutations of the phenotype data. Then, we can proceed with FDR correction by ranking all the nominal $P$ values in increasing order and by counting how many $P$ values in the permuted data sets are smaller. This immediately gives an FDR estimate: if we have 500 $P$ values in the permuted data sets being smaller than the 100th smallest nominal $P$ value, we can then assume that the FDR for the 100 first associations is around 5% ($=500/(100 \times 100)$).

(2) *Approximate permutation scheme*. To enable fast screening in *trans*, we also designed an approximation of the method described just above based on what we already do in *cis*. To make it possible, we assume that the phenotypes are independent and normally distributed (which can be enforced in practice). Then, we draw from the null by permuting only one randomly chosen phenotype, testing for associations with all variants in *trans* and storing the smallest $P$ value. When we repeat this many times (typically 1,000 or 10,000 times), we effectively build a null distribution of the strongest associations for a single phenotype. We then make it continuous by fitting a beta distribution as we do in *cis* and use it to adjust every nominal $P$ value coming from the initial pass for the number of variants being tested. To correct for the number of phenotypes being tested, we estimate FDR (using $R/q$ value) again as we do in *cis*; that is onto the best adjusted $P$ values per phenotype (one per phenotype). As a by-product, this also gives an adjusted $P$ value threshold that we finally use to identify all phenotype-variant pairs that are whole-genome significant. In our experiments, this approach gives similar results to the full permutation scheme both in term of FDR estimates and number of discoveries (Supplementary Fig. 15).

**Integrating molQTLs with functional annotations.** QTLtools includes two approaches to integrate molQTLs with functional annotations. First, it can measure the density of functional annotations around the genomic positions of molQTLs. To do so, we first enumerate all annotations within a given window around the molQTLs (by default 1 Mb). Then, we split this window into small bins (default 1 kb) and count the number of functional annotations overlapping each bin. This produces an annotation count per bin that can be then plotted to see if there is any peak or depletion around the molQTLs (Fig. 2e). Complementary to this density-based representation, QTLtools can also assess if the molQTLs overlap the functional annotations more often than what we expect by chance. Here, we mean by chance what is expected given the non-uniform distributions of molQTLs and functional annotations around the genomic positions of the molecular phenotypes. To do so, we first enumerate all the functional annotations located nearby (for example, within 1 Mb) a given molecular phenotype. In practice, for $X$ phenotypes being quantified, we have $X$ lists of annotations. And, for the subset $Y$ of those having a significant molQTL, we count how often the $Y$ molQTLs overlap the annotations in the corresponding lists: this gives the observed overlap frequency $fobs(Y)$ between molQTLs and functional annotations. Then, we permute randomly many times (typically a 1,000 times) the lists of functional annotations across the phenotypes (for example, phenotype $A$ may be assigned the list of annotations coming from phenotype $B$) and for each permuted data set, we count how often the $Y$ molQTLs do overlap the newly assigned functional annotations: this gives the expected overlap frequency $fexp(Y)$ between molQTLs and functional annotations. By doing this permutation scheme, we keep unchanged the distribution of functional annotations and molQTLs around molecular phenotypes. Now that we have the observed and expected overlap frequencies, we use a fisher test to assess how $fobs(Y)$ and $fexp(Y)$ differ. This gives an odd ratio estimate and a tow-sided $P$ value which basically tells us first if there is enrichment or depletion and second how significant this is. Then, we typically plot these two quantities on a scatter plot with the $x$ axis and $y$ axis being the odd ratio and the significance of the enrichment/depletion, respectively (Fig. 2f). In our experiments, we use three types of functional annotations generated by ENCODE[12] for lymphoblastoid cell lines: open chromatin regions given by DNAse footprinting, a union of all transcription factor binding sites assayed by ChIP-seq and transcribed regions as predicted by ChromHMM[21].

**Data availability.** The Geuvadis RNA-seq data corresponds exactly to what has been generated in the original Geuvadis study, so please consult the Supplementary Materials of the paper[1] for a more detailed description of the experimental protocol used for RNA-seq data generation. In our experiments, we focus our attention on a subset of 358 European samples for which we also have complete DNA sequence data generated as part of the phase 3 of the 1,000 Genomes project[22]. All variant sites with a minor allele frequency across all 358 samples below 5% or exhibiting more than two possible alleles have been removed which resulted in a set of 6,241,929 single-nucleotide variants and 843,851 short insertion–deletions or structural variants left for the analysis. All the raw sequence data can be downloaded from http://www.ebi.ac.uk/arrayexpress/files/E-GEUV-1/processed/ (RNA-seq data) and ftp://ftp.1000genomes.ebi.ac.uk/vol1/ftp/release/20130502/ (for DNA-seq data)

The histone modification data set contains ChIP-seq across for 3 histone modifications across 47 European samples: H3K4me1, H3K4me3 and H3K27ac that are known to usually tag enhancers, promoters and active regions. Please consult this paper[7] for more detailed description of the experimental protocols used for the ChIP-seq data generation. In this data set, the samples have been either

sequenced or imputed from an Illumina OMNI2.5 M as part of the phase 1 of the 1,000 Genomes project[11]. Again, all variant sites with a minor allele frequency across the 47 samples below 5% or exhibiting more than two possible alleles have been removed which resulted in a set of 6,085,881 single-nucleotide variants and 606,344 short insertion–deletions or structural variants. All the raw ChIP-seq data can be downloaded from https://www.ebi.ac.uk/arrayexpress/experiments/E-MTAB-3657/.

QTLtools is open source and available for download at https://qtltools.github.io/qtltools/.

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

## Acknowledgements

This research is supported by grants from European Commission SYSCOL FP7, European Research Council, Louis Jeantet Foundation, Swiss National Science Foundation, SystemsX, the NIH-NIMH (GTEx) and Helse Sør Øst. The computations were performed at the Vital-IT Swiss Institute of Bioinformatics.

## Author contributions

O.D., H.O., A.A.B. and E.T.D. designed the research. O.D. and H.O. implemented the methods. O.D. analysed data. A.F. and N.I.P. helped to test various functionalities. O.D. and E.T.D. supervised the research. O.D. wrote the paper.

## Additional information

**Competing interests:** The authors declare no competing financial interests.

**Publisher's note**: 

