## [Peer Review File · Nature Communications]

Reviewers' comments:

Reviewer #1 (Remarks to the Author):

In this paper the authors describe a set of software tools for performing eQTL analysis. In principle the tools should work with any molecular QTLs (e.g. DNA methylation or histone modification QTLs), however the main focus of the paper and the current version of the tools is on eQTLs and RNA-seq data.

The QTLTools software provides many functions that are part of common pipelines for eQTL analysis, including QC, quantification, PCA, cis-eQTL discovery. It also provides some less-common features of standard pipelines such as trans-eQTL discovery, enrichment of functional annotations, and overlap with GWAS hits.

A main selling point of the QTLTools pipeline is that it is very fast, and thus will be useful when working with large datasets or when trying to call trans-eQTLs. A lot of the speedup is accomplished by approximating the null permutation distribution using a Beta distribution, as published in their earlier FastQTL paper. This makes it possible for the authors to quickly perform conditional analyses, trans-eQTL discovery etc.

The code is well-written and is quite well-documented with examples provided on the website. Many researchers will benefit from the availability of this fast, standardized pipeline.

My criticisms of the paper are mostly minor:

1. Most of the methodological detail is in the supplemental material. The paper would be easier to follow if some of this material could be moved to the main text (if page and figure limits allow).
2. Several places in the manuscript refer to genes that are "quantified" or not quantified: For example, Line 57: "get 22,147 genes quantified in more than half the samples" The meaning of this is not entirely clear. A gene or exon is still "quantified" even if it has low or 0 expression so long as it can be measured. I think this would be better described as exceeding a minimum expression threshold in 50% of samples. From Supplemental Figure 4, the threshold used appears to be $> 50\%$ of samples having $\log_{10}(\text{RPKM}) > 0$. The legend has a mistake in that it says $\text{RPKM}=0$, rather than $\log_{10}(\text{RPKM})>0$.
3. The link to the VCF index file on several of the online documentation pages is missing: http://jungle.unige.ch/QTLtools_examples/genotypes.chr22.vcf.gz.csi
4. The authors cite many papers from their group, but hardly any QTL papers from other groups.
5. In the supplement, the data from the paper by Waszak et al. is referred to as the "Sinergia data set" in a few places. This seems like a strange (possibly internal working name) for this dataset. Why not just "histone modification dataset" or ChIP-seq data from Waszak et al.?
6. There are a very large number of grammatical and spelling errors throughout the paper and supplement. If possible, the paper should be edited by a native English speaker to correct these. Here are a few of them (there are many more):

Author Affiliations:

"Denetic Medicine" => "Genetic Medicine"

Abstract:

"become a standard" => "have become a standard"

"This kind of datasets" => "These kind of datasets"

"well established" => "well-established"

Line 112: "and then standard FDR methods" => "and then use standard FDR methods"

Line 125: "to which extend" => "to which extent"

Line 127: "functional annotations as those" => "functional annotations such as those"

Supplement Line 27: "supplementary materials the paper" => "supplementary materials of the paper"

"ChIP-seq" is mis-capitalized as "ChiP-seq" in several locations in the supplement

Storey & Tibshirani is misspelled Storey-Tarabishi in the supplement.

Reviewer #2 (Remarks to the Author):

The current manuscript presents a new command line tool, QTLtools, C++ based to analyse large "omic" data sets. The tool aims at tackling three main challenges in the field of omic data analysis: Efficient False Discovery rate control, various type of datasets handling and ability to compute these efficiently in terms of time and accuracy.

If not new, these challenges are the major ones faced by the research community to deliver pertinent results in large scale meta-analysis in the medical field. The presented tool offers several answers to the addressed challenges, notably by offering a statistical framework for multiple testing in the context of expression data and by streamlining the analytical process from raw dataset to interpretable results. Beyond the human genomic research community, it may raise interest of others such as animal or plant genomics.

The presented work shows innovative solutions in the field and is very well documented. The manuscript does suffer of majors flaws and is well written.

However it could improved by briefly listing other high throughput tools available (i.e. plinkseq <https://atgu.mgh.harvard.edu/plinkseq/> , dissect <http://www.dissect.ed.ac.uk/>, GWAMA <http://www.geenivaramu.ee/en/tools/gwama> or METAL <http://csg.sph.umich.edu/abecasis/metal/>) and explain briefly to the reader –a potential user- in which way QTLtools complement the aforementioned tools in some cases and/or perform innovative and efficient detection methods regarding these tools, notably the cis/trans analysis.

Regarding the large p small n challenge that the authors aim to address, 358 individuals from the GEUVADIS study are tested (I.41), knowing the large amount of tissues from the GTEx, this remain a lower population size as what can be now encountered in meta-GWAS studies as underlined by the authors I. 26-28. Therefore, simulation study including a larger set of individuals would be welcome to assess the scalability of the presented tool.

A strong asset of the proposed framework is the result visualization offered by including core R functionalities in the QTLtools package release. Installing and testing QTLtools functionalities, we notice the absence of the script repository providing most of the figures. We assume this is an oversight from the authors and should be easy to correct. To make a full use of the proposed tool it

has to be available and therefore is part of revisions to get the manuscript accepted.

REVIEWERS' COMMENTS:

Reviewer #1 (Remarks to the Author):

The authors have addressed my concerns.

One minor comment: Figure 2F x-axis should be "Odds ratio" not "Odd ratio"

Reviewer #2 (Remarks to the Author):

The comments and suggestions made in the previous report regarding the full availability online of the R code and clarifications regarding multiple testing were addressed by the authors.

Additionally, the supplementary data has been reformatted to ease the reading of the manuscript.

Finally the new supplemental Figure 8 clarifies the running time for the complete GTEx data, composed of independent tissue datasets.

A simulation study of tool's scalability regarding computational performance on much larger set of individuals (>1000s) tested would have been welcome.

However, this lack does not affect the quality of the study, nor tool's innovative character.

This manuscript should be accepted for publication.

Reviewer #1 (Remarks to the Author):

In this paper the authors describe a set of software tools for performing eQTL analysis. In principle the tools should work with any molecular QTLs (e.g. DNA methylation or histone modification QTLs), however the main focus of the paper and the current version of the tools is on eQTLs and RNA-seq data.

The QTLTools software provides many functions that are part of common pipelines for eQTL analysis, including QC, quantification, PCA, cis-eQTL discovery. It also provides some less-common features of standard pipelines such as trans-eQTL discovery, enrichment of functional annotations, and overlap with GWAS hits.

A main selling point of the QTLTools pipeline is that it is very fast, and thus will be useful when working with large datasets or when trying to call trans-eQTLs. A lot of the speedup is accomplished by approximating the null permutation distribution using a Beta distribution, as published in their earlier FastQTL paper. This makes it possible for the authors to quickly perform conditional analyses, trans-eQTL discovery etc.

The code is well-written and is quite well-documented with examples provided on the website. Many researchers will benefit from the availability of this fast, standardized pipeline.

We thank the reviewer for this positive comment.

My criticisms of the paper are mostly minor:

1. Most of the methodological detail is in the supplemental material. The paper would be easier to follow if some of this material could be moved to the main text (if page and figure limits allow).

We thank the reviewer for this useful suggestion and we proceeded accordingly. We moved substantial fraction of the supplementary in the main text. Specifically, we created a new methods section that describes the main functionalities of QTLtools and moved three figures from the supplementary in the main paper. In addition, we also merge all supplementary material into a unique document. We believe the paper to be clearer now and to fit better the nature communications format requirements.

2. Several places in the manuscript refer to genes that are “quantified” or not quantified: For example, Line 57: “get 22,147 genes quantified in more than half the samples” The meaning of this is not entirely clear. A gene or exon is still “quantified” even if it has low or 0 expression so long as it can be measured. I think this would be better described as exceeding a minimum expression threshold in 50% of samples. From Supplemental Figure 4, the threshold used appears to be $> 50\%$ of samples having $\log_{10}(\text{RPKM}) > 0$. The legend has a mistake in that it says $\text{RPKM}=0$, rather than $\log_{10}(\text{RPKM})>0$.

The reviewer is right about gene quantifications so we corrected this in all three documents. However, the legend of supplementary figure 4 is actually correct. The blue bars indeed show the distribution of genes with less than 50% of the samples having null quantifications (RPKM=0).

3. The link to the VCF index file on several of the online documentation pages is missing: http://jungle.unige.ch/QTLtools_examples/genotypes.chr22.vcf.gz.csi

We fixed these links on the webpage. Thanks for spotting this.

4. The authors cite many papers from their group, but hardly any QTL papers from other groups.

We added two additional references to key eQTL studies in the field.

5. In the supplement, the data from the paper by Waszak et al. is referred to as the "Sinergia data set" in a few places. This seems like a strange (possibly internal working name) for this dataset. Why not just "histone modification dataset" or ChIP-seq data from Waszak et al.?

The reviewer is right; it was an internal name so we now refer to it as the histone modification dataset and we changed all documents accordingly.

6. There are a very large number of grammatical and spelling errors throughout the paper and supplement. If possible, the paper should be edited by a native english speaker to correct these. Here are a few of them (there are many more):

Author Affiliations:

"Denetic Medicine" => "Genetic Medicine"

Abstact:

"become a standard" => "have become a standard"

"This kind of datasets" => "These kind of datasets"

"well established" => "well-established"

Line 112: "and then standard FDR methods" => "and then use standard FDR methods"

Line 125: "to which extend" => "to which extent"

Line 127: "functional annotations as those" => "functional annotations such as those"

Supplement Line 27: "supplementary materials the paper" => "supplementary materials of the paper"

"ChIP-seq" is mis-capitalized as "ChiP-seq" in several locations in the supplement

Storey & Tibshirani is misspelled Storey-Tarabishi in the supplement.

We corrected all the spelling errors mentioned above and asked to a native English speaker to further correct the main manuscript which resulted in additional corrections.

Reviewer #2 (Remarks to the Author):

The current manuscript presents a new command line tool, QTLtools, C++ based to analyse large “omic” data sets. The tool aims at tackling three main challenges in the field of omic data analysis:

Efficient False Discovery rate control, various type of datasets handling and ability to compute these efficiently in terms of time and accuracy.

If not new, these challenges are the major ones faced by the research community to deliver pertinent results in large scale meta-analysis in the medical field. The presented tool offers several answers to the addressed challenges, notably by offering a statistical framework for multiple testing in the context of expression data and by streamlining the analytical process from raw dataset to interpretable results. Beyond the human genomic research community, it may raise interest of others such as animal or plant genomics.

The presented work shows innovative solutions in the field and is very well documented. The manuscript does suffer of majors flaws and is well written.

We thank the reviewer for this positive comment.

However it could improved by briefly listing other high throughput tools available (i.e. plinkseq <https://atgu.mgh.harvard.edu/plinkseq/> , dissect <http://www.dissect.ed.ac.uk/>, GWAMA <http://www.geenivaramu.ee/en/tools/gwama> or METAL <http://csg.sph.umich.edu/abecasis/metal/>) and explain briefly to the reader –a potential user- in which way QTLtools complement the aforementioned tools in some cases and/or perform innovative and efficient detection methods regarding these tools, notably the cis/trans analysis.

We understand the point of the reviewer. We slightly modified the manuscript and added two references to account for this in the introduction.

Regarding the large p small n challenge that the authors aim to address, 358 individuals from the GEUVADIS study are tested (l.41), knowing the large amount of tissues from the GTEx, this remain a lower population size as what can be now encountered in meta-GWAS studies as underlined by the authors l. 26-28. Therefore, simulation study including a larger set of individuals would be welcome to assess the scalability of the presented tool.

We added a new figure in the supplementary materials (Supplementary Figure 8) that shows the running times required to map eQTLs in cis for all 44 tissues of the last GTEx study as a function of the sample size for each tissue. This shows the linear scaling of the QTL mapper and gives a good idea of the time needed to process up to 7,000 samples (=870 CPU hours).

A strong asset of the proposed framework is the result visualization offered by including core R functionalities in the QTLtools package release. Installing and testing QTLtools functionalities, we notice the absence of the script repository providing most of the figures. We assume this is an oversight from the authors and should be easy to correct. To make a full use of the proposed tool it has to be available and therefore is part of revisions to get the manuscript accepted.

This is a fair point. We therefore packaged as much as we could all the R scripts and data required to reproduce pretty much all the figures of the paper. This can now be found on the webpage (<https://qtltools.github.io/qtltools/>): there is a link on the left panel called "QTLtools figures".